# Associations of the Lipidome with Ageing, Cognitive Decline and Exercise Behaviours

**DOI:** 10.3390/metabo12090822

**Published:** 2022-08-31

**Authors:** Maria Kadyrov, Luke Whiley, Belinda Brown, Kirk I. Erickson, Elaine Holmes

**Affiliations:** 1Australian National Phenome Centre, Health Futures Institute, Murdoch University, Harry Perkins Building, 5 Robin Warren Drive, Murdoch, WA 6150, Australia; 2Centre for Computational and Systems Medicine, Health Futures Institute, Murdoch University, Harry Perkins Building, 5 Robin Warren Drive, Murdoch, WA 6150, Australia; 3Discipline of Exercise Science, College of Science, Health, Engineering and Education, Murdoch University, 90 South Street, Murdoch, WA 6150, Australia; 4Centre for Healthy Ageing, Health Futures Institute, Murdoch University, 90 South Street, Murdoch, WA 6150, Australia; 5Perron Institute for Neurological and Translational Science, Nedlands, WA 6009, Australia; 6School of Medical Sciences, Sarich Neuroscience Research Institute, Edith Cowan University, Nedlands, WA 6009, Australia; 7Department of Psychology, University of Pittsburgh, Pittsburgh, PA 15260, USA; 8AdventHealth Research Institute, Neuroscience Institute, Orlando, FL 32804, USA; 9PROFITH “PROmoting FITness and Health Through Physical Activity” Research Group, Sport and Health University Research Institute (iMUDS), Department of Physical Education and Sports, Faculty of Sport Sciences, University of Granada, 18071 Granada, Spain; 10Division of Integrative Systems and Digestive Medicine, Department of Surgery and Cancer, Imperial College London, London SW7 2AZ, UK

**Keywords:** lipidomics, exercise, ageing, cognition, metabolism, metabolic phenotyping, liquid chromatography–mass spectrometry (LC–MS), nuclear magnetic resonance spectroscopy (NMR)

## Abstract

One of the most recognisable features of ageing is a decline in brain health and cognitive dysfunction, which is associated with perturbations to regular lipid homeostasis. Although ageing is the largest risk factor for several neurodegenerative diseases such as dementia, a loss in cognitive function is commonly observed in adults over the age of 65. Despite the prevalence of normal age-related cognitive decline, there is a lack of effective methods to improve the health of the ageing brain. In light of this, exercise has shown promise for positively influencing neurocognitive health and associated lipid profiles. This review summarises age-related changes in several lipid classes that are found in the brain, including fatty acyls, glycerolipids, phospholipids, sphingolipids and sterols, and explores the consequences of age-associated pathological cognitive decline on these lipid classes. Evidence of the positive effects of exercise on the affected lipid profiles are also discussed to highlight the potential for exercise to be used therapeutically to mitigate age-related changes to lipid metabolism and prevent cognitive decline in later life.

## 1. Introduction

Ageing is an inevitable natural phenomenon and is associated with alterations in various biological processes leading to the gradual decline of physiological functions [1]. With a global increase in the ageing population, older adults (60+ years) are expected to outnumber young children (<9 years) by 2030 [2]. By 2050, the proportion of people over the age of 60 is expected to double to 2 billion (22% of the projected population) [3]. This increase in population size and consequent healthcare demands and retirement living costs imposes substantial socioeconomic pressure on communities and healthcare systems. In Australia alone, the projected cost of ageing to the economy is AUD 36 billion by 2028 [4]. Elucidating ageing processes and highlighting new ways of prolonging health will not only have a substantial economic impact, but will also relieve the pressures on healthcare systems and have a profound benefit at the individual and societal level. 

One of the most recognisable features of ageing is a decline in brain health, commonly manifesting in brain atrophy and cognitive decline [5]. As such, ageing is the largest risk factor for neurodegenerative diseases such as dementia [6]. Dementia is an umbrella term that describes a collection of symptoms including cognitive impairment, memory loss and behavioural changes observed in various neuropathological disorders [7]. There are many conditions that can cause dementia; however, it is most commonly a result of Alzheimer’s disease (AD) [8]. Hence, determining ways of preserving the health of the ageing brain is a global priority. 

A common misconception about ageing is that a decline in brain health is solely observed in those who have dementia. In fact, a loss in cognitive function and a reduction in brain volume is commonly observed in most adults over the age of 65 [9,10]. Despite the prevalence of normal age-related cognitive and brain volume changes, there is a lack of effective methods for improving the health of the ageing brain. In light of this, exercise has shown promise for influencing neurocognitive health [11] (Figure 1). There is, however, inconsistent evidence regarding the effectiveness of exercise in maintaining a healthy ageing brain. Some studies report clear exercise-related improvements in brain health and cognition in older adults [12], whereas others report very small effect sizes [5] or no effect at all [13]. The lack of consensus about the benefits of exercise on neurocognitive health may be one reason why the prescription of exercise for improving brain health is not widely adopted. More research is vital to understanding exercise parameters that contribute to the greatest cognitive response, as well as to gain understanding of individual variability in exercise-induced brain benefits [14,15,16]. Whilst previous research has primarily focused on characterising the relationships between exercise, cognitive function and brain volume, far less research has been carried out on elucidating the biological mechanisms that are involved in these associations, particularly on a lipidomic level.

Traditional “reductionist” approaches, which divide explanations of physiological behaviour into separate components, have facilitated noteworthy progress in elucidating the mechanisms underlying exercise and healthy brain ageing. These approaches, however, have been limited to studying a particular molecule, for example cholesterol, in isolation. Although they have added to our knowledge of how exercise influences brain health, there is still a poor understanding of how integrative lipid networks underlie this relationship. Fortunately, the growing field of the “omics” sciences, in particular lipidomics, has provided new opportunities to better understand the complex and interconnected nature of lipid networks in the context of ageing and age-related disease and to evaluate the impact of exercise on the ageing brain. Lipidomics, a subcategory of metabolomics, involves the comprehensive analysis of lipids in biological fluids, allowing for the biochemical interrogation of the physiological state of an individual [20]. The lipidome, which is the collection of all lipids within a biological system, can vary greatly from one system to another; e.g., the plasma lipidome will be vastly different to the cerebrospinal fluid (CSF) lipidome. Currently, it has been estimated that there are over 200,000 unique lipid species in the human body, making it a rich source of biological information [21]. Perturbations to the lipidome can be interrogated with the use of spectroscopic platforms consisting of high resolution nuclear magnetic resonance (NMR) spectroscopy (measuring lipoproteins) and mass spectrometry (MS) measuring specific lipid classes and species, coupled with multivariate statistical methods [22]. Interrogation of NMR and MS spectra of biofluids containing information on thousands of lipid compounds can give unique insights into in vivo cellular processes [23]. To date, most lipidomic studies in this field have focused on identifying biomarkers of age-related diseases for diagnostic and prognostic purposes. Few studies have investigated the lipidomic profiles of “healthy agers” and explored the association of these lipids with cognitive function. By understanding lipidomic profiles of “healthy agers”, the mechanisms underlying their successful ageing may be uncovered and a quantifiable phenotype indicative of good brain health may be established. Furthermore, the availability of this information opens the opportunity for monitoring of the efficacy of exercise interventions in preventing or slowing age-related cognitive decline.

By understanding age-related processes and elucidating the underlying biochemical changes that occur in the brain, new opportunities for early intervention to prevent or delay physiological decline may be identified. In particular, new knowledge of brain ageing mechanisms that are reflected in the lipidome may facilitate implementation of exercise interventions aimed at beneficial lipid modulation. Here, this review paper will discuss specific lipid classes that are associated with cognitive function and ageing and highlight evidence that exercise may be used to mitigate these changes to prevent age-related cognitive decline. Identification of these lipid profiles may be used to assess the effectiveness of future exercise interventions aimed at improving cognitive function in older adults.

## 2. Does Exercise Influence Neurocognitive Health?

The functional capability of the brain progressively declines with increasing age [9]. Cognitive abilities can be categorised into several domains such as attention, language, memory, visuospatial abilities and executive functions, all of which experience quantifiable declines after the age of 60 [24]. The slowing of cognitive processing, another age-related change to cognition, can directly contribute to several other domains such as motor coordination, learning, decision making and attention [25]. Lifestyle factors such as physical activity and exercise have shown promising neurocognitive health benefits throughout life [14]. Indeed, physical inactivity has been identified as one of the greatest modifiable risk factors for unhealthy brain ageing [26] and has been shown to contribute to a third of all dementia cases [27]. When reviewing the literature, it is important to define the difference between physical activity and exercise. Physical activity is defined as any movement of skeletal muscles that results in energy expenditure (e.g., household activities, sports, occupational activities), whereas exercise is a subset of physical activity that is planned, structured and repetitive, and undertaken with the aim of improving or maintaining fitness [28].

Numerous cross-sectional analyses of cohort studies have provided reports of enhanced cognitive functioning and reduced cognitive decline associated with high levels of physical activity [29]. In particular, domains such as global cognition [30,31], executive function [32], attention [33] and verbal memory [30] appear to gain the greatest benefit from higher physical activity levels. However, the relationship between physical activity and cognition requires investigation beyond cohort studies, as most use data from self-reported physical activity surveys and questionnaires, which may be prone to biases and may exclude consideration of confounders such as the social impact of some forms of exercise. It is worth mentioning that some of the benefits of physical activity may be social and psychological rather than physical (i.e., increased participation in group activities may counteract the impact of loneliness); hence, it can be difficult to measure the impact of increased physical activity in isolation. Furthermore, it can be difficult to determine causality in observational cohort studies, as there may be additional variables that account for associations between physical activity and cognition that remain unknown. Nevertheless, the results from these studies have provided valuable proof-of-concept evidence of the relationship between exercise and cognition, paving the way for randomised controlled trials (RCT) to further establish the relationship between exercise and cognition.

Unlike the literature on observational studies, which predominantly concludes that physical activity or exercise has a positive effect on cognition, results from RCTs have not been as consistent. Numerous meta-analyses and systematic reviews of RCTs have concluded that exercise has a significant benefit on cognition [5,34,35,36], whilst others have reported insufficient statistical power to detect an effect of exercise on cognition [14,37,38]. In fact, a 2015 Cochrane review [13] concluded that there was not sufficient evidence to suggest aerobic exercise contributes to cognitive benefit in older adults. However, an umbrella review of 76 meta-analyses and systematic reviews conducted by Erickson et al. [36] for the 2018 Physical Activity Guidelines Advisory Committee Scientific Report [39] determined that there was moderate-to-strong evidence for physical activity benefiting cognitive function at all stages of the lifespan. Another review also reported that physical activity improved cognitive function regardless of cognitive status[34].

The inconsistent results emerging from exercise RCTs may be attributed to the vast interindividual variability in cognitive response to identical exercise programs [15,16,40]. As such, studies have begun to categorise individuals from within the same exercise groups as “responders” and “non-responders” according to whether they showed benefit or not [41]. The identification of “responders” to exercise suggests that interventions may need to be tailored to characteristics of the individual. As with the biological mechanisms that underpin the association between healthy brain ageing and exercise, the differences between “responders” and “non-responders” have not been well studied and require further investigation. A powerful approach to investigating heterogeneity in individual response to intrinsic factors (i.e., ageing) and extrinsic stimuli (i.e., exercise) is lipidomics, which can profile a wide range of lipids and characterise specific panels of lipids associated with physiological or pathological states.

## 3. Lipidomics

### 3.1. Why Lipids?

Lipids are a diverse group of organic compounds which comprise non-polar hydrocarbon chains that are essential molecules for healthy brain structure and function [42]. With lipids making up half of its dry weight, the brain has one of the largest concentrations of lipids in the body, second only to adipose tissue [43,44]. The lipids most abundantly found in the brain can be classified into five main classes: fatty acyls (FA), glycerolipids (GL), glycerophospholipids (GP), sphingolipids (SP) and sterols (ST) (Figure 2) [45], each of which play a role in maintaining brain health by maintaining the structural integrity of membranes, cellular signalling and energy metabolism [46]. There are many benefits to studying lipids in the context of brain ageing. Firstly, lipids are the downstream products of all other facets of biological regulation, making the investigation of the lipidome an attractive subject area for expanding the knowledge of complex biological systems in the context of brain ageing [47]. This means that the lipidome reflects all the collective changes that have occurred during processes stemming from the genome, transcriptome and proteome [48,49]. The lipidome is also sensitive to external environmental influences, thereby providing the most proximal representation of an individual phenotype [50,51]. In other words, while the genome, transcriptome and proteome may provide an inference of what may happen in a biological system, the lipidome can provide a “snapshot” of what is currently occurring [52,53]. This is due to the rapid nature of lipid turnover, which occurs in seconds, unlike proteins or mRNA transcripts, which may take minutes or hours to respond to a stimulus such as exercise [53]. In addition, lipidomic analysis requires very small sample volumes (ranging from 10–50 μL for MS, and 150–500 μL for NMR) to detect, identify and quantify compounds, which in turn makes sample collection relatively easy [53]. Finally, the detection of changes in previously unidentified lipids is possible with untargeted lipidomics methods, allowing for hypothesis generation to uncover novel mechanisms related to disease pathogenesis [54,55].

### 3.2. Measurement of the Lipidome

There are several combinations of platforms available for use in lipidomic approaches, of which the most commonly used is MS [56,57]. A highly sensitive and specific analytical tool, MS is used to measure the mass-to-charge ratio (m/z) of molecules in a sample [56,57]. The high sensitivity of MS can enable rich coverage of the lipidome, as it can detect lipids in the nM to pM range [58]. MS is commonly coupled with various chromatographic techniques, such as liquid chromatography (LC) [59]. Together, the resolution, sensitivity and selectivity of LC–MS instruments combined can allow for the identification of isobaric lipids that have the same elemental composition, but different chemical structure [60]. Chromatographic separation can also provide additional information on the physiochemical properties of analytes by measuring the retention times. This separation prior to MS detection offers greater analyte resolution, which is particularly useful for mixtures that are complex in nature, such as biological fluids [59,60]. Despite these advantages to MS, there are drawbacks, such as the need for extensive sample preparation prior to analysis [56]. Furthermore, lipids must first be ionised in order to be detected by an MS instrument. Hence, regardless of the sensitivity, if a lipid does not ionise, then it is undetectable. Furthermore, quantitation is not a standard feature of untargeted MS, although addition of labelled chemical standards can be used to achieve quantitative results and species-level identification of lipids. Processing times for sample preparation and data acquisition with MS can be quite long, in the order of hours [56]. Furthermore, MS platforms do not provide atom-centred data and give limited information on molecular mobility, which can be obtained from NMR spectroscopy.

Complementary to MS, NMR spectroscopy is based on the interaction of atomic nuclei with radio-waves, typically in the presence of a powerful magnet, which allows for the identification of an atom and can provide information on its relative location within a lipid [56,61,62]. There are many advantages to using NMR spectroscopy in lipidomics, including the fact that it is a highly reproducible technique (typically > 98%) [63] and is inherently quantitative [56]. Absolute quantification of CE, GL, GP, cholesterol and unsaturated fatty acids can be determined with ^1^H NMR experiments; however, since lipids typically contain long hydrocarbon side chains, characterisation of lipid species is unattainable due to significant signal overlap from the hydrogen atoms. NMR-based measurement of lipoproteins involves minimal sample preparation, allowing for the integrity of the sample to be preserved [64], and does not require the removal of high abundance molecules, such as proteins, prior to analysis. As such, NMR can be used to readily quantify lipoproteins, which is a challenging and labour-intensive task with MS and hence is not commonly performed in MS lipidomics assays [65]. Furthermore, NMR can generate information on the molecular environment and mobility of lipids and lipoproteins [65]. This can be used to identify unknown lipids, which is especially useful when investigating novel biomarkers [64]. However, the overall sensitivity of NMR in comparison to other analytical approaches can be much lower, which can result in important molecules being overlooked [61,62]. Furthermore, substantial sample volumes (up to 500 µL) are required. Potential biomarkers that are present in trace amounts may be missed during analysis, as the presence of compounds in very high concentrations can often obscure compounds that are present in lower concentrations [66]. As neither MS nor NMR methods alone can provide a truly comprehensive assessment of the entire lipidome, it is common to see studies incorporating the use of both techniques in parallel.

### 3.3. Challenges with Lipidomics

Although there are many advantages to studying lipids, it does not come without its limitations. Lack of standardisation, due to the rapid advancement of lipidomics, has resulted in inconsistencies in methods, analytical platforms and how lipids are reported [67]. Lipids are immensely diverse in their structure and complexity (e.g., lipid isomers and isobars); hence, it is difficult to have a single uniform analytical platform that can detect all lipids [68,69]. This vast structural diversity of lipids makes it very challenging to develop and implement chemical standards for the identification and quantitation of all lipid species. The lack of labelled commercial standards makes lipid annotation and identification very difficult, resulting in investigators having to resort to reporting the “sum composition”, rather than a more appropriate “structurally defined molecular lipid” [69,70]. In glycerophospholipids, for example, there are five key factors that are required for a lipid to be annotated at the “structurally defined molecular lipid” level [69]. These include:(1)Characterisation of the lipid head group (e.g., phosphocholine (PC));(2)The length of the carbon chains (e.g., C12);(3)The enantiomeric configuration of linkages at the sn-1 and sn-2 positions on the glycerol backbone (e.g., acyl, alkyl or alkenyl);(4)The number, location and stereochemistry of any C=C bonds, or “unsaturated” bonds (e.g., 18:2 (9Z, 12Z) is a chain of 18 carbons with two C=C bonds at the ninth and twelfth carbon positions);(5)Any occurrences of modifications within these chains (e.g., 20:1(5Z)-OH(12S) is a chain of 20 carbons, with a C=C bond at the fifth carbon and a hydroxyl group on the twelfth carbon).

When all of this information is not available, a lipid such as PC (16:1/18:3) may be oversimplified and noted as PC 34:4, as is the case for many lipids reported in the literature, thus losing significant functionally relevant data. This then makes it difficult to compare findings across studies that report lipids using different notations. For instance, it cannot be assumed that all PC 34:4 lipids are PC (16:1/18:3), as they may also be a PC (16:2/18:2) or PC (18:3/16:1). This lack of comparability can result in many biologically relevant findings being overlooked; e.g., two independent studies may an find an identical lipid that is associated with cognitive decline, but without the detailed “structurally defined molecular lipid” level notation, one may report their lipid as PC 34:0 and another as PC 16:0/18:0, rendering it difficult to compare the findings of both studies.

## 4. Impact of Ageing, Cognitive Dysfunction and Exercise on the Lipidome

During ageing, substantial changes to lipid metabolism occur that gradually diminish physiological functions, such as genomic damage, increased oxidative stress and impaired mitochondrial function leading to perturbed energy metabolism [71]. In fact, it has been shown that starting at around 50 years of age, up to 14% of the entire brain lipidome is impacted by ageing [72]. Structural changes are also a common feature of brain ageing, including changes to membrane lipid composition [73,74] and brain hypotrophy[10]. These changes in lipid metabolism ultimately result in impairments to cognitive function and can increase vulnerability to neurodegenerative diseases, specifically those that are age-related such as Alzheimer’s disease [74]. As most instances of pathological cognitive decline occur slowly during the prodromal stages of disease development [75,76], it can be hard to determine which of these are normal age-associated changes and which are early diagnostic signs of neurodegenerative diseases. The following section summarises changes in different classes of lipids in association with ageing, cognitive decline and exercise.

### 4.1. Fatty Acyls

#### 4.1.1. Fatty Acids

Fatty acids are carboxylic acids comprised of long aliphatic chains that are either saturated, containing no C=C double bonds, or are unsaturated, with one or more C=C double bonds [77]. Whilst most fatty acids can be synthesised endogenously, essential fatty acids must be obtained from the diet, as they cannot be synthesised in sufficient quantities by the human body [78]. These essential fatty acids are classified into omega-6 (n-6) or omega-3 (n-3) polyunsaturated fatty acids, or PUFAs. Linoleic acid, the parent n-6 fatty acid, is metabolised to produce arachidonic acid (AA), while the parent omega-3 fatty acid, alpha-linolenic acid (ALA), is metabolised to form docosahexaenoic acid (DHA) and eicosapentaenoic acid (EPA) [79]. Phospholipids in the brain have been shown to contain high quantities of long-chain PUFAs, which is an indication of their importance for healthy nervous system function. These PUFAs all have crucial roles in maintaining the fluidity of cell membranes and ion channel activity in neurons, endothelial cells and glia. DHA and AA can also stimulate the increase of acetylcholine, a major neurotransmitter that can influence cognitive function by enhancing synaptic plasticity. PUFAs have a major role in signalling, as they are the precursors for oxylipins, which are potential chemical messengers that are critical in immune and inflammatory regulation. The most common of these are a family of lipid mediators called eicosanoids that are derived from AA. As such, AA metabolism is often involved in proinflammatory pathways, whereas EPA and DHA have more anti-inflammatory effects. EPA and AA are especially important, as EPA works by competing with AA to reduce pro-inflammatory molecules that are induced by AA [80]. Changes in fatty acyls with ageing, cognitive decline and exercise are well characterised in the literature and summarised below in Table 1.

##### Fatty Acids in Ageing

Several SFAs, MUFAs and PUFAs have been shown to increase in plasma with age [95,106,114]. In a study of the adult human plasma metabolome, Lawton et al. [106] identified four fatty acids that were elevated in older adults, including stearic acid, oleic acid, linoleic acid and arachidonic acid. Stearic acid was also associated with ageing in plasma in a recent study by Xie et al. [114]. Other studies show that ageing had varied effects on plasma fatty acids, as described by Menni et al. [95], who reported increased levels of heptadecenoic acid and EPA with age, but a decrease in eicosadienoic acid. In contrast, Jové et al. [118] observed age related reductions in plasma levels of EPA. Changes to PUFA levels in CSF have also been described. In a study of 114 healthy Japanese participants, Saito et al. [119] identified ten lipids that were positively associated with age, eight of which contained either eicosatrienoic acid, EPA or DHA. On the other hand, PUFAs such as DHA and AA have been shown to progressively decline with age in the orbitofrontal cortex region of the brain [121]. The authors suggested that this decline in PUFAs may contribute to the atrophy observed in the grey matter volume that occurs with increasing age. This is due to the key roles that PUFAs have in stimulating neuronal activity, synaptogenesis and neurogenesis, whilst also protecting against neuroinflammation and apoptosis [126]. Additionally, Carver et al. [127] showed that AA and DHA also decrease with age in the cerebral cortex and erythrocytes, whilst linoleic acid increases. They also reported a significant association between brain and erythrocyte levels, particularly for palmitic acid, suggesting that some erythrocyte fatty acid levels may be useful in predicting fatty acid levels within the brain. Further evidence of this is reported by Goozee et al. [111], who reported a decrease in DHA levels in erythrocytes, which coincides with the reduced levels observed in the brain [121,127]. It is clear that ageing has varied effects on fatty acids in the central nervous system (CNS) and the periphery, which may be an indication of disturbed fatty acid transport through the blood–brain barrier (BBB). Disturbances to fatty acid, particularly PUFA, availability to neurons are detrimental, as they can cause mitochondrial dysfunction, oxidative damage and changes to membrane fluidity [128].

##### Fatty Acids and Cognition

Perturbed fatty acid metabolism has also been implicated in cognitive dysfunction. In animal studies, there is extensive evidence that DHA depletion in the brain can lead to learning and memory problems [129,130,131] and that a deficiency in dietary omega-3 PUFAs impairs cognitive performance by negatively impacting dopamine metabolism [132]. Cognitive decline has also been linked to decreased levels of PUFAs in humans. For example, Tan et al. [133] analysed erythrocyte levels of EPA and DHA in 1575 cognitively normal older adults. They reported that low DHA concentrations were associated with poorer performance in abstract thinking, visual memory and executive function. Furthermore, participants in the lowest quartile of DHA concentrations had a lower total brain volume and showed more white matter hyperintensities (indicator of small-vessel vascular disease [134]). Similarly, Bigornia et al. [117] reported that increased erythrocyte eicosadienoic acid levels were associated with lower Mini-Mental State Examination (MMSE) and executive function scores, whilst elevated baseline erythrocyte AA levels were predictive of cognitive impairment at a two year follow up. Cross sectional analysis of a longitudinal cohort of 2251 older adults by Beydoun et al. [112] found that the risk of global cognitive decline was increased with elevated plasma levels of palmitic acid and AA, and reduced levels of linoleic acid, EPA and DHA. Reduced DHA has also been associated with declines in memory and executive function in cognitively normal older adults, and more rapid cognitive decline in individuals with AD [123]. In apparent contradiction to studies that report a positive association, Palacios et al. [120] found that elevated EPA levels in plasma were associated with worse cognitive function. Discrepancies in results may be due to differences in PUFA quantification, particularly regarding the lipid fractions that were measured. This was demonstrated by Kou et al. [135], who showed that while there were no reductions in total brain DHA levels, a significant decrease in DHA was observed when only looking at the plasmalogen fractions.

##### Fatty Acids and Exercise

There is a large body of evidence that demonstrates that exercise increases circulating levels of fatty acids. This is due to exercise-induced increases in lipolytic activity and the subsequent release of stored fatty acids, such as DHA and EPA, from triacylglycerols in adipose tissue. These free fatty acids, also referred to as non-esterified fatty acids (NEFA), are crucial energy sources for working muscles; however, when they are utilized is dependent on exercise intensity. Plasma NEFAs are typically elevated during moderate intensity exercise (>50% VO_2max_) [136,137], and progressively increase with the duration of exercise. This was demonstrated in a study by Watt et al. [138], who reported elevated plasma NEFAs after 90 min of cycling at 57% VO_2max_ compared with levels at rest, which continued to increase progressively until cessation (240 min). As exercise intensity increases (>70–80% VO_2max)_, NEFA mobilisation begins to decline as muscle glycogen takes over as the main fuel source utilized by active muscles [139].

Whilst exercised-induced elevations in circulating NEFAs and their role in energy metabolism for working muscles has been well studied, the mechanism by which these NEFAs benefit the brain are still not well understood. One possible mechanism may be via brain-derived neurotrophic factor, or BDNF, a neurotrophin that is released with exercise. BDNF has shown to have a major role in supporting neuronal plasticity, particularly in the memory centres of the brain such as the hippocampus [140]. It is widely reported that exercise increases BDNF in plasma and in the central nervous system. In 2011, Erickson et al. [141] demonstrated that serum levels of BDNF were increased with exercise, and this was associated with an increase in hippocampal volume and improvements in memory. Interestingly, DHA—which can be increased through fat oxidation of DHA-carrying triacylglycerols—has also been demonstrated to promote the release of BDNF [142]. Whilst a relationship is evident with BDNF, there is some disagreement in the literature about the mechanisms by which exercise and DHA influence cognition. Some studies, such as that of Chytrova et al. [143], suggest that DHA is the underlying factor that enables neuronal growth, elevated synaptic plasticity and improved cognitive function, and that exercise purely enhances this mechanism. On the other hand, studies such as that of Wu et al. [142] claim that the opposite is true and, in fact, exercise is the driving force behind improvements in cognition and synaptic plasticity, and that the presence of DHA enhances this effect. Although evidence suggests that there is a synergistic action between exercise and PUFAs on cognitive function, further research is required to understand these mechanisms.

#### 4.1.2. Acylcarnitines

Acylcarnitines are criticial for the transport of fatty acids to mitochondrial cells for beta oxidation [144]. Once fatty acids are synthesised, they are often combined with glycerol and are stored as triacylglycerols as energy reserves. When the breakdown of these triacylglycerols (i.e., lipolysis) occurs, the unbound or “free fatty acids” are released into the blood stream. They are then carried by plasma albumin, as they are insoluble on their own, and are transported to the mitochondria [145]. Whilst short- and medium-chain fatty acids can easily diffuse through the inner mitochondrial membrane, long-chain fatty acids rely on carnitine for transportation. Carnitines also have a key role in transporting toxic compounds out of the mitochondria and preventing the accumulation of fatty acids, both of which can be deleterious to the health of the cell if left unchecked [146].

##### Acylcarnitines in Ageing

Changes to circulating acylcarnitine levels in older adults have been well described in the literature, as highlighted in Table 1. Most acylcarnitines have been associated with an increase in concentration with ageing, particularly long-chain and very-long-chain acylcarnitines. This may be an indication of impaired mitochondrial function, as elevated levels of acylcarnitines occur as a result of reduced fatty acid oxidation in the mitochondria. Furthermore, a recent study reported that plasma acylcarnitine levels were correlated with plasma neurofilament light chain protein (NFL) protein, a marker of neurodegeneration [100]. Increases in acylcarnitines have also been implicated in age-related diseases [90] and increases in proinflammatory signalling [147]. Of the 24 acylcarnitines summarised in Table 1, only two were reduced in association with aging. The two acylcarnitines that were reduced—pimelylcarnitine and nonanoylcarnitine—are both acylcarnitines that carry odd-chain fatty acids (OCFA). OCFAs are primarily found in dietary sources such as milk [148]; hence, this reduction in OCFA carrying acylcarnitines may reflect a decrease in the amount of diary consumed or a reduced capacity for absorption of dairy products, rather than being due to advancing age.

##### Acylcarnitines and Cognition

Whilst high concentrations of acylcarnitines in plasma can be indicative of mitochondrial dysfunction, diminished acylcarnitine levels can also be deleterious, as they reflect impaired transport of fatty acids to the mitochondria and subsequent reduced energy metabolism [149]. This is what has been observed in several studies investigating circulating acylcarnitine concentrations in individuals with varying levels of cognitive impairment. Cristofano et al. [84] measured serum levels of 34 acylcarnitines in individuals with normal cognition, subjective memory complaint (SMC), mild cognitive impairment and AD. The authors reported a progressive decrease in several acylcarnitines, including acetylcarnitine, malonyl-, 3-hydroxyisovaleryl-, C6:0-, C10:0-, C12:0-, C12:1-, C14:0-, C14:1-, C16:1-, C18:0-, C18:1- and C18:2-L-carnitine, which decreased in order of increasing cognitive impairment (CN > SMC > MCI > AD). This suggests that acylcarnitines are gradually diminished from cognitively normal to cognitive impairment, and that metabolism of these acylcarnitine species is closely connected to cognitive health. Reduced levels of several acylcarnitines have also been associated with worse cognition and lower MMSE scores in AD [85]. Furthermore, higher baseline levels of malonyl-, hexenoyl-, pimelyl-, and tetradecenoyl-carnitine are associated with reduced AD risk and slower decline in global cognition, executive function, episodic memory and semantic memory in cognitively healthy individuals and those with MCI [94]. A recent study by Nho et al. [150] reported that elevated levels of propionylcarnitine were associated with higher memory scores and reduced amyloid accumulation in MCI and AD.

##### Acylcarnitines and Exercise

Coinciding with the literature on the effects of exercise on fatty acids, studies unanimously show that exercise acutely increases circulating levels of acylcarnitines [86,87,88,89,93,97,109]. As previously mentioned, exercise stimulates lipolysis to release fatty acids which act as an energy source for working muscles [151]. As carnitine is required to transport long chain fatty acids (LCFAs) across the inner mitochondrial membrane for beta oxidation, an increase in acylcarnitines is also observed [146]. This in turn increases the capacity for energy metabolism and clearance of toxic waste products, which is vital for organs such as the brain. With exercise, the acute increase in acylcarnitines may help to counteract the negative effects of perturbed acylcarnitine metabolism that is observed with ageing and cognitive decline. As diminished concentrations are observed with cognitive decline, there may be the potential for exercise-induced increases in acylcarnitines to mitigate and replenish these levels to alleviate, prevent or delay further age-related cognitive decline.

#### 4.1.3. Ketone Bodies

In addition to changes in the lipidome with age, it is also worth considering the role of ketone bodies, as they are lipid-derived molecules. Ketone bodies are the end products of fatty acid metabolism, and are the main alternative fuel source for the brain [152]. They are produced when there is an increased energy demand, low glucose availability and elevated fatty acids, such as when fasting or exercising [153]. As the brain has a huge energy demand, requiring about 20% of the total energy expenditure of the body, a constant source of energy is essential for normal brain function [154]. This is especially important in age-related neurological conditions in which normal brain glucose metabolism may be impaired, which is a common feature in several age-associated neurodegenerative diseases including AD, Parkinson’s disease, Huntington’s disease (HD) and amyotrophic lateral sclerosis (ALS) [155]. In these diseases, glucose hypometabolism is often correlated with disease severity; therefore, supporting the brain’s energy requirements through alternative fuel sources such as ketone bodies may slow the progression of disease [156].

##### Ketone Bodies in Ageing

One of the most abundant ketone bodies is beta-hydroxybutyrate, or β-HB. There is growing evidence that increased β-HB has “anti-ageing” properties because, apart from its role in brain energy metabolism, β-HB is also involved in several cell signalling functions [157]. This was demonstrated in a study by Lawton et al. [106] that showed that β-HB was elevated in a cohort of “healthy agers”. Whilst there are many studies that investigate the effects of β-HB supplementation, fasting or ketogenic diets on ketone levels in older adults, there is a gap in the literature characterising the changes that occur with advancing age; hence, future research is needed in this area.

##### Ketone Bodies and Cognition

Whilst there is no published evidence that we are aware of regarding the association between ketone bodies with cognitive performance in healthy participants, they have been associated with cognition in cases of neurodegenerative disease. Circulating β-HB has been shown to be decreased in AD patients [103], leading to suggestions that increasing β-HB may alleviate symptoms of cognitive decline. This was demonstrated in a study by Fortier et al. [107], who showed that elevated levels in plasma ketone bodies were positively associated with improvements in cognitive function in individuals with MCI. Dietary ketones have long been used to elevate ketone levels, and recent work has revealed that AD patients on a ketogenic diet showed non-significant improvements in cognition and significant improvement in quality of life and daily function, two important factors for those living with dementia [158].

##### Ketone Bodies and Exercise

As ketones are produced in situations of increased energy demand, circulating levels are significantly elevated with exercise acylcarnitines [86,93,97]. As previously mentioned, BDNF is increased with exercise, inducing a beneficial response in cognition due to enhanced synaptic plasticity. Sleiman et al. [159] demonstrated that this is achieved through the release of β-HB during exercise. The authors showed that exercise induced increases in β-HB, which stimulated the activity of promotors of the BDNF gene, upregulating gene expression and BDNF production. Studies that explore the relationship between exercise-induced β-HB and cognitive function are limited, and further research is required to determine if exercise can induce the anti-ageing effects of β-HB and improve age-related cognitive outcomes.

### 4.2. Glycerolipids

Glycerolipids are comprised of fatty acids that are attached to a glycerol backbone via ester linkages [160]. They can have up to three fatty acids, and are termed monoacylglycerol (MAG), diacylglycerol (DAG) and triacylglycerol (TAG), respectively. Glycerolipids are crucial energy reservoirs; if there is excess glucose in the body, it will be converted to TAG for storage [161]. High blood TAG levels are an indicator of elevated blood glucose levels, a hallmark of metabolic disturbances and insulin resistance [162]. Glycerolipids are also important signalling molecules in the CNS [163]. One of the most well studied MAGs, 2-arachidonoylglycerol (2-AG), is an endocannabinoid neuromodulator that has a crucial role in appetite control, immune system regulation and pain [164]. DAG is also very important, as it is an activator of protein kinase C (PKC), an enzyme that is essential for modulation of membrane excitability and the release of neurotransmitters via the phosphorylation of tau protein [165]. Disturbances to normal tau phosphorylation have been associated with neurodegenerative diseases such as Alzheimer’s disease [166]. TAG also has a crucial role in brain signalling as it can cross the BBB and interfere with leptin receptors in the hypothalamus, the hormone that regulates hunger, and insulin, which controls blood sugar levels [167]. This can be detrimental, as dysregulated leptin and insulin metabolism can increase hunger and fat storage, thus leading to metabolic diseases such as obesity, type-II diabetes and cardiovascular disease, each of which are risk factors for age-related neurodegenerative diseases [167]. Changes in glycerolipids with ageing, cognitive decline and exercise are well characterised in the literature and summarised below in Table 2.

#### 4.2.1. Glycerolipids in Ageing

Advancing age has been shown to impact glycerolipid metabolism, leading to increased risk of metabolic disease [170]. Fanelli et al. [168] reported an increase in 2-AG in obese pre-menopausal and lean post-menopausal women, suggesting that in women, ageing is associated with a disturbance in 2-AG metabolism that is comparable to the metabolic dysfunctions observed in obesity. In animal studies, a significant reduction in DAG and TAG was observed in the amygdala of aged mice, the brain region responsible for emotional memory and motivation [171]. Similarly, several DAG species have been shown to be reduced in plasma [100,118,172], serum [173] and CSF [174] in humans. In contrast, TAG concentrations are typically elevated with ageing [173], which may be due to slower clearance rates of TAG. A recent study found that serum levels of three TAG species, TAG 48:1, TAG 50:0 and TAG 52:1, were significantly higher in older adults than in their younger counterparts [170]. This supports the argument that ageing may be associated with an increased risk of metabolic disease, as studies have reported that serum levels of TAGs containing LCFAs are linked to an increased risk of cardiovascular disease [175]. Elevated TAG levels have also been linked to constant, low-level inflammation that occurs with ageing without any presence of infection [176]. As inflammation is associated with perturbed adipose tissue function, there is a crucial link between altered glycerolipid metabolism and ageing. In CSF, changes in TAGs with ageing have been varied, with some reporting an increase in concentrations [119,173,174], while others report a decrease [100,101].

#### 4.2.2. Glycerolipids and Cognition

The relationship between cognition and glycerolipids has been extensively studied, but with conflicting findings. Olazaran et al. [98] demonstrated that plasma TAG levels progressively decreased from participants with normal cognition to MCI to AD, suggesting that lower levels of TAGs were associated with increasing severity of cognitive impairment, whereas others report no association between cognition and TAG levels. This was demonstrated in a study by Huang et al. [177], who investigated the relationship between cognitive impairment and triacylglycerol levels in the serum of Chinese nonagenarians (90+ years old) and centenarians (100+ years old). The authors reported no significant differences between serum TAG levels in participants with cognitive impairment compared to cognitively normal controls. In contrast, most studies in the literature have suggested that glycerolipid concentrations have an inverse relationship with cognition, as high TAG levels have been associated with poor recall and verbal knowledge in humans and lower memory and learning scores in mice [167]. In fact, two glycerolipid species (triacylglycerol 50:1 and diacylglycerol 18:1_18:1) showed such a strong correlation with the extent of brain atrophy that when combined with MMSE (mini mental state examination) results, they increased the reliability of MCI diagnosis. Whilst many studies show that high plasma TAG is negatively associated with cognition, Yin et al. [178] paradoxically reported the opposite. In their cross-sectional study of 836 individuals aged 80 and above, high plasma triacylglycerol levels were associated with higher MMSE scores (better cognition). Hence, the authors suggested that high plasma TAG may have a role in preserving cognitive function in adults over the age of 80. Whilst there are conflicting findings in the associations between glycerolipids and cognitive impairment, the evidence suggests that perturbed glycerolipid metabolism in involved in age-related cognitive decline in some capacity.

#### 4.2.3. Glycerolipids and Exercise

Although no specific glycerolipid species showed changes in all three topics of interest (ageing, cognitive decline and exercise), there was a clear pattern in the effects of exercise on other glycerolipids. Karl et al. [86] observed the effects of exercise in military personnel and reported a decrease in 12 diacylglycerol species, five of which were also reduced with exercise in a recent study by Contrepois et al. [169]. These included DAG 16:0/18:1, 16:0/18:2, 16:1/18:2, 18:1/18:1 and 18:1/18:2. Studies have also shown that exercise influences plasma TAG levels; however, this change appears to be specific to the duration and extent of exercise and can be varied amongst different individuals. Mougios et al. [179] demonstrated that plasma triacylglycerols decreased in a group of 19 males playing two 30 min games of handball with a 10 min rest interval between games. While most players showed a decrease in the first game, followed by an even greater decrease in the second, five players showed an increase in the second half. The authors suggested that while exercise increases lipolysis of triacylglycerols to free fatty acids for energy metabolism, the liver may counteract this by releasing more triacylglycerols, explaining the increase observed in some players.

### 4.3. Glycerophospholipids

As key components of the lipid bilayer in cell membranes, glycerophospholipids are one of the most abundantly occurring lipid classes in the brain [180]. Also referred to as phospholipids, glycerophospholipids are similar to glycerolipids in that they contain up to two fatty acids attached to a glycerol backbone, but differ by the addition of a polar head group [181]. These polar groups are what define each subclass of phospholipids, and can be a choline, serine, inositol or ethanolamine, forming phosphatidylcholine (PC), phosphatidylserine (PS), phosphatidylinositol (PI) and phosphatidylethanolamine (PE), respectively [182].

Phosphatidylcholine is the most abundant phospholipid found in the plasma membrane, and is also used in the brain to produce acetylcholine, one of the body’s key neurotransmitters [183]. In the peripheral nervous system, acetylcholine is the primary neurotransmitter that regulates activation of skeletal muscles and autonomic bodily functions, such as digestion [184]. In the CNS, acetylcholine functions as both a neurotransmitter and a neuromodulator in cholinergic areas involved in attention, memory, motivation and arousal [185]. Cholinergic dysfunction can have serious consequences to brain health and has been associated with cognitive decline that occurs with Alzheimer’s disease [186]. Whilst treatment options are limited, some of the available pharmaceutical therapeutics for AD target this deficiency and increase levels of acetylcholine to alleviate some of the cognitive symptoms that occur [187].

Phosphatidylserine (PS) is mainly found on the inner plasma membrane and is highly enriched in myelin [188]. Accounting for 40% of all PUFAs in the brain [189], DHA is mostly stored as PS; hence, it is thought that PS in cell membranes acts as a reservoir for DHA [190]. PE is also found on the inner plasma membrane and predominantly carries arachidonic acid [191]. Phospholipids that contain acyl chains with multiple unsaturated bonds, such as arachidonic acid, are more “fluidic” than those that contain saturated fatty acyl chains [192]. As such, increases in PEs can increase the fluidity of a cell membrane, which in turn has been shown to regulate various cellular processes and signalling pathways, both physiologically and pathologically [193]. PI also characteristically contain arachidonic acid, as well as stearic acid [194]. Making up only 10% of the total phospholipids, PIs are the least abundant. However, they and their metabolites have crucial roles in regulating membrane signalling and transport, lipid homeostasis and membrane biogenesis [195]. Changes in phospholipids with ageing, cognitive decline and exercise are well characterised in the literature and summarised below in Table 3.

#### 4.3.1. Glycerophospholipids in Ageing

Glycerophospholipids are one of the most well studied lipid classes in ageing. Recently, Verri Hernandes et al. [82] investigated the serum metabolome of 6872 participants to identify age associated metabolites. They reported 73 phospholipids that were related to age, with only PC ae 44:5 declining with advancing age. This was also described by Rist et al. [199]; however, it was only significant in women. Reduced levels of several other PC, PE and LysoPC species have also been reported in plasma [100,118,172] and serum [196]. A recent lipidomics study of “successful agers” by Montoliu et al. [173] compared the lipid profiles of centenarians with elderly participants that were below the age of 100 (average age 70.4 ± 6) and revealed that 23 phospholipid species (3 PIs, 5 PEs, 15 PCs) differed between the two groups. All but two of these lipids were found in higher concentrations in centenarians, which, interestingly, were both PCs that contained saturated fatty acyl chains. Due to their presence in the profiles of centenarians, the authors suggested that these lipids were modulators of healthy ageing and are associated with longevity. On the other hand, age-associated changes to phospholipids in CSF have been varied, with some reporting an increase [119,174] and others a decline [200].

#### 4.3.2. Glycerophospholipids and Cognition

Studies investigating phospholipid metabolism in cognitively normal older adults are limited. However, altered phospholipid metabolism has repeatedly been associated with neurodegenerative disease. Whiley et al. [201] identified three PC species (PC 16:0/20:5, 16:0/22:6 and 18:0/22:6) that showed a reducing trend from cognitively normal controls to MCI and AD. Interestingly, these PCs all either carried an EPA or a DHA fatty acyl chain, two of the most important PUFAs for healthy brain function. In fact, similar downward trends in levels of PCs with increasing degrees of cognitive impairment were also reported in several other studies [92,98,104]. This downward trend suggests that reductions in these PC species are positively associated with the severity of cognitive impairment and that they may be involved in the mechanisms that underpin this decline in cognitive function.

#### 4.3.3. Glycerophospholipids and Exercise

As increased levels of PUFA-carrying PCs are associated with healthy ageing, and reduced concentrations are observed in age-related cognitive decline, strategies to increase PUFA-carrying PCs may have a protective effect on the ageing brain. Exercise may be one such strategy, as there is evidence that exercise impacts phospholipid metabolism. Whilst many phospholipid species have been implicated with ageing, cognitive decline and exercise independently, only one was consistently changed across all three. PC 18:1/20:4, containing an AA acyl chain, was elevated with ageing in CSF [174], reduced in AD serum [197] and increased with exercise [169]. Exercise-induced elevations in PC 18:1/20:4 could potentially have a positive effect on cognition, as they may counteract the effects of cognitive impairment that occur with the diminished levels of PC 18:1/20:4 in AD. The effects of exercise-induced elevations in PUFA-containing PCs on cognitive function are not well understood, and further research is needed in this area.

### 4.4. Sphingolipids

Unlike all other lipid classes, sphingolipids have a sphingosine backbone rather than glycerol [202]. They are critical for brain development, as they are important for efficient signal transduction, membrane structure and cell recognition [42]. As neurons are polarised cells, they rely on the selective transportation of molecules through the cellular membrane to ensure efficient synaptic transmission and neuronal connectivity [203]. This regulation of neural homeostasis is achieved via sphingolipids, and as a result, they are highly enriched in the cellular membranes of neurons [204]. Key sphingolipids include ceramides and sphingomyelins. Ceramides are the precursor compounds for the synthesis of more complex sphingolipids and have an important signalling role in cell proliferation, differentiation and apoptosis [205]. Sphingomyelins are a major component of myelin, the insulating material that surrounds axons that allow efficient transmission of electrical impulses, and oligodendrocytes, which are the glial cells that produce myelin and provide metabolic nourishment to axons [206]. Changes in sphingolipids with ageing, cognitive decline and exercise are well characterised in the literature and summarised below in Table 4.

#### 4.4.1. Sphingolipids in Ageing

Age-related changes to sphingolipid metabolism can have a strong impact on brain function; however, these changes are varied and are species specific. In a study of 6055 twins, Menni et al. [95] reported an increase in SM 18:1/14:0 and SM 18:1/16:0 in the plasma of older adults in comparison to their younger counterparts. Increases in palmitic acid (C16:0) have been linked to immune dysregulation, as it has been shown to trigger the secretion of tumour necrosis factor-a (TNF-a) and interleukin-6, two pro-inflammatory cytokines, and the activation of caspase-3 (apoptosis) [209]. Elevated sphingolipids, particularly ceramides, in muscle tissue have also been linked to insulin resistance, which may contribute to age-associated declines in insulin sensitivity. In contrast, Chatterjee et al. [100] also reported reductions in several SM species, including SM 39:1, SM 41:1 and SM 42:1, in the plasma of cognitively normal older adults. For another SM species, Lim et al. [101] showed that SM 18:1/22:0 was reduced in adults over the age of 60. Using data from the Framingham Offspring Cohort, McBurney et al. [210] found that older adults with higher concentrations of behenic acid (C22:0) lived longer than those with lower levels.

#### 4.4.2. Sphingolipids and Cognition

Studies investigating sphingolipid metabolism in cognitively normal older adults are limited. However, altered sphingolipid metabolism has repeatedly been associated with neurodegenerative disease. Olazaran et al. [98] reported a progressive decrease in plasma levels of SM 39:1, SM 41:1 and SM 42:1 from cognitively normal older adults to individuals with MCI and AD. This suggests that diminished levels of these SM species are associated with the degree of cognitive decline observed in these age-related conditions. Fote et al. also observed a decline in SM 18:1/22:0 and SM 18:1/16:0 in the CSF of patients with MCI [207]. Interestingly, other research groups have observed an increase in several SM species in MCI serum, which then declined in AD. In fact, this same trend has also been reported in several ceramide species, suggesting that there may be a link between increased levels of certain sphingomyelins and ceramides and subsequent cognitive decline.

#### 4.4.3. Sphingolipids and Exercise

The effects of exercise on sphingolipids are varied and specific to the exercise type and lipid subclass. Chronic endurance exercise training has been demonstrated to decrease concentrations of ceramides in muscle tissue [211], whereas after a single bout of exercise, an increase in muscle ceramides was observed. Increased ceramides have also been reported in serum, as described by Bergman et al. [212], who found elevated levels after a single bout of exercise (90 min at 50% VO_2max_), which then returned to baseline levels during recovery. Interestingly, serum sphingosine 1-phosphate and sphingomyelins showed no change during exercise, but showed diminished levels following cessation, suggesting that they may have a role in recovery [212]. Another study by Saleem et al. [213] investigated the association between cardiopulmonary fitness and sphingolipids in older adults with coronary artery disease, or CAD. Individuals with CAD are at risk of developing cognitive impairment due to impaired cerebral blood flow perfusion and disruptions to the blood–brain barrier [214]. The authors found that elevated baseline levels of SM 18:1, CER 16:0, CER 18:0, CER 20:0 and CER 24:1 were associated with poorer VO_2peak_, and that improvements in cardiopulmonary fitness were associated with reductions in SM 18:1, CER 16:0, CER 18:0 and C24:1. Studies have also shown positive effects of exercise on age- and cognition-associated lipids, as demonstrated by Varga et al. [208], who reported an increase in serum levels of SM 41:1, a sphingolipid that was reported to decrease with age and cognitive decline. Similarly, Karl et al. [86] also demonstrated elevated levels of several sphingomyelin species following exercise, many of which are impaired with ageing and cognitive decline. This provides evidence that exercise can modulate detrimental sphingolipid profiles, highlighting the importance of beneficial lifestyle factors in older adults to mitigate age-related changes to sphingolipid metabolism to support healthy brain ageing and better cognitive function.

### 4.5. Sterols

#### 4.5.1. Cholesterol

Sterols are a class of lipids that contain a steroid group [215]. The most abundant sterol lipid in animals is cholesterol, which is a precursor for several hormones and fat-soluble vitamins and is also as a key component of cell membranes [216]. Whilst only accounting for 2% of the weight of the human body, the brain contains a quarter of the total cholesterol, with most of it residing in myelin [217]. Cholesterol homeostasis in the CNS is vastly different compared to other body systems. Outside of the CNS, cellular requirements for cholesterol are provided through the diet or via de novo synthesis in the liver. This cholesterol is then bound to apolipoproteins to form lipoproteins, and are carried through the blood for uptake at the cell surface [217]. Some examples include low-density lipoproteins (LDL), which carry cholesterol to tissues, and high-density lipoproteins (HDL), which carry cholesterol away from tissues for breakdown at the liver [218]. These two classes of lipoproteins are colloquially referred to as “bad” and “good” cholesterol, respectively, although the reality is almost certainly more complex. Uptake at the cell surface is not feasible for the brain, as lipoproteins cannot cross the blood–brain barrier; hence, most of the cholesterol in the brain is synthesised de novo within the CNS [219]. Once synthesised, the cholesterol is then transported around the CNS to neurons via apolipoprotein-E (APOE), the brain’s main cholesterol carrier [219]. As previously mentioned, carriers of the APOE ε4 allele are at an increased risk of developing AD [220]. This is due to the detrimental effects of impaired APOE metabolism on brain health, including aggregation of the neurotoxic amyloid-beta peptides, disruption to BBB function and increased inflammation [221]. Changes in sterols with ageing, cognitive decline and exercise are well characterised in the literature and summarised below in Table 5.

##### Cholesterol in Ageing

Changes to cholesterol metabolism are a prominent feature of ageing and are sex dependent. In men, concentrations of cholesterol in plasma increase gradually with age until about 50 years of age, whereupon they start to decrease. In women however, Rist et al. [199] showed that cholesterol was elevated in older, post-menopausal women when compared to their younger counterparts. This is due to the drop in oestrogen levels that occur during menopause, as oestrogen helps to regulate lipid metabolism in the liver, leading to an increase in LDL levels. Cholesterol metabolism in the brain has also been shown to decline with age, with decreases in cholesterol synthesis being reported in the hippocampus [224]. Whilst some suggest that reduced cholesterol turnover is protective as it helps to keep cholesterol levels stable during ageing [225], others say that this may contribute to cognitive decline, as impaired cholesterol synthesis has been linked to reduced synaptic plasticity [226]. Furthermore, the integrity of the BBB has shown to decline with age, which may impair its ability to keep peripheral cholesterol separate from the CNS [227]. In mice, Saeed et al. [228] showed that BBB breakdown resulted in the entry of cholesterol from the blood into the brain. Additionally, age-associated declines in microglial function in processing excess cholesterol leads to an accumulation of cholesterol in microglia [229]. This, in turn, induces inflammatory responses and prevents remyelination of myelin sheets encompassing neurons.

##### Cholesterol and Cognition

Due to the cholesterol-rich nature of the brain, any changes to cholesterol metabolism can lead to impairments in brain structure and function. Whilst brain cholesterol is separate from circulating cholesterol, there still appears to be a link between circulating levels of cholesterol and cognitive function. Yu et al. [99] demonstrated that higher serum cholesterol levels in midlife were positively associated with a more rapid decline in global cognitive function in later life. In contrast, another study showed that higher levels of serum cholesterol in women in mid- and late-life were predictive of slower processing speed and reduced episodic memory [230].

##### Cholesterol and Exercise

The benefits of exercise in the context of cholesterol metabolism have been widely studied. A meta-analysis of 51 studies that included 12 or more weeks of aerobic exercise was conducted by Leon and Sanchez [231]. The authors found that, on average, prolonged aerobic exercise had no effect on total cholesterol, but considerable improvements to the HDL:LDL cholesterol ratio were observed due to a 4.6% increase in HDL and 5% reduction in LDL cholesterol. An increase in HDL cholesterol levels was also observed in patients with AD following a 16-week moderate-to-high intensity exercise intervention, compared to non-exercise AD controls. This suggests that exercise may be beneficial to regulating cholesterol homeostasis in AD patients, and future research is required to see whether this improvement in cholesterol profiles has an effect to cognitive function.

#### 4.5.2. Steroid Hormones

##### Steroid Hormones and Ageing

Hormone changes are also a major element of ageing, particularly involving sex hormones. Age-related declines in plasma testosterone are well recognised, particularly after the age of 40 [232]. Several longitudinal studies have demonstrated a gradual decrease in plasma testosterone with advancing age in healthy men [233,234]. In women, changes in sex hormones are more pronounced, as characterised by the dramatic decline in oestrogen and progesterone levels during menopause [235]. Both testosterone and oestrogen are produced from dehydroepiandrosterone (DHEA) and its sulphated counterpart (DHEA-S) [223]. There is also evidence that DHEA and DHEA-S are involved in the modulation of various processes in the brain and have neuroprotective capabilities due to their roles as antioxidants. Steady declines in the levels of DHEA-S were shown to be associated with increasing age in several studies in both men and women; however, the total levels of DHEA-S in men were significantly higher than in women [236]. In fact, when comparing DHEA plasma levels of 50–60-year-old adults to 20–30-year-olds, Labrie et al. [237] found a 74% and 70% decrease in DHEA levels in serum of men and women, respectively.

##### Steroid Hormones and Cognition

As with levels of circulating steroid hormones, the pattern of their release may also be significantly altered with ageing. Perturbations to steroid hormones under circadian regulations such as the sleep wake cycle can result in poor neurocognitive health [238]. Diminished sleep quality in older adults have been linked to declines in executive function and attention [239]. Testosterone has also shown to decline with age in men and has been linked to cognitive health [240]. Older men with higher levels of testosterone have been reported to have better cognitive function than those with lower levels [241]. Men with diminished levels of testosterone showed improvements to cognition following supplementation of testosterone. While there is evidence that testosterone is linked to cognition in older men, the exact mechanism by which this occurs is unclear.

In women, oestrogen decline has been linked to cognitive impairments. Oestrogen has several roles in supporting neurocognitive health [242]. In the hippocampus and prefrontal cortex, two crucial regions involved in cognitive processes, oestrogen has a role in regulating synapse formation and turnover [243]. Oestrogen also promotes the synthesis of neurotrophins [244], modulates dopaminergic, serotonergic and cholinergic neurotransmitter pathways [245], and protects the brain against inflammation and stress [246]. Studies have suggested that rapid declines in oestrogen that are observed in menopausal women may render the brain vulnerable to neurodegeneration [247]. This may have a role in the increased risk of women developing cognitive decline in later life, as women with dementia outnumber men twofold [248].

##### Steroid Hormones and Exercise

The effects of exercise on steroid hormones are specific to the type of exercise and the fitness of the individual. A study by Sato et al. [249] compared male elite athletes with untrained age matched controls who underwent two exercise sessions consisting of 15 min at 40% VO_2peak_, 70% VO_2peak_ and then 90% VO_2peak_ until exhaustion, with 10 min of rest between each interval. Exercise at all intensities induced elevated serum free testosterone and DHEA in the non-trained group, whereas these were only elevated at 90% VO_2peak_ in the elite athlete group. This suggests that serum sex hormone changes in response to exercise are influenced by exercise intensity in individuals with varying levels of physical fitness. As cardiorespiratory fitness declines with age, this is promising, as there is evidence that older adults may still benefit from the effects of exercise on steroid hormone regulation.

## 5. Conclusions

Increased propensity for healthy ageing will contribute to enhanced socioeconomic and personal outcomes, yet there is currently a lack of consistent evidence for promotion of brain health in older adults. Whilst exercise has been shown to be effective in regulating healthy lipid profiles, there are numerous gaps in the knowledge of how these benefits translate to the ageing brain and whether they may produce a quantifiable improvement in cognitive health. Intrinsic factors (such as genetics) and extrinsic influences (such as diet) likely contribute to brain ageing outcomes; hence, lipidomics has emerged as a promising tool to explore these relationships. Although lipidomics has provided valuable insights on the effects of age and exercise on the lipidome, research on age-associated lipid responses to exercise is lacking. This is due to most of the exercise lipidomics papers being focused on human performance; as such, they frequently involve the study of elite athletes. Whilst this research is very important, these findings may not be transferable to the average older adult, as they may elicit a vastly different lipidomic response to exercise. Further research is also required to determine the lipidomic characteristics that distinguish between individuals who respond well to exercise and show significant neurocognitive benefits and those who do not.

## Figures and Tables

**Figure 1 metabolites-12-00822-f001:**
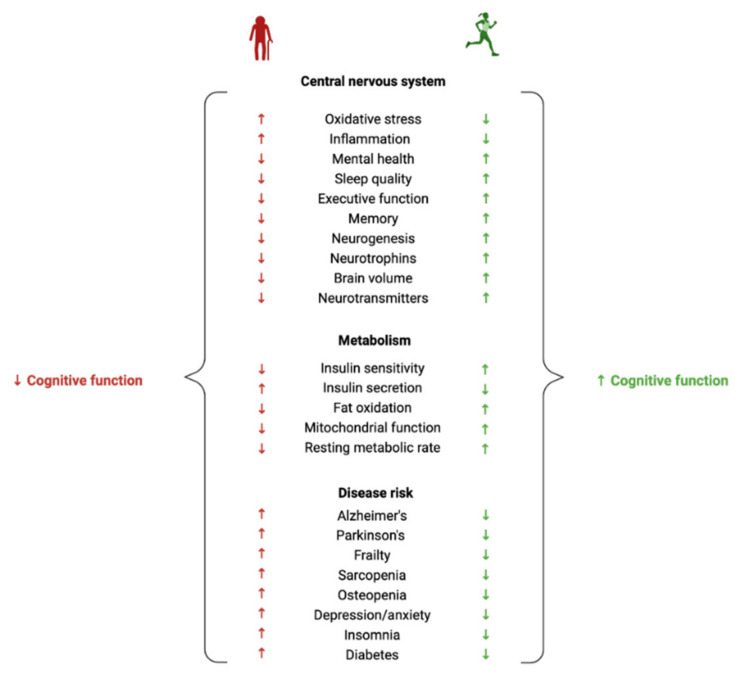
Effects of exercise on age-associated physiological changes. See Chodzko-Zajko et al. [17], Garatachea et al. [18] and Carapeto and Aguayo-Mazzucato [19] for comprehensive reviews of the anti-ageing effects of exercise.

**Figure 2 metabolites-12-00822-f002:**
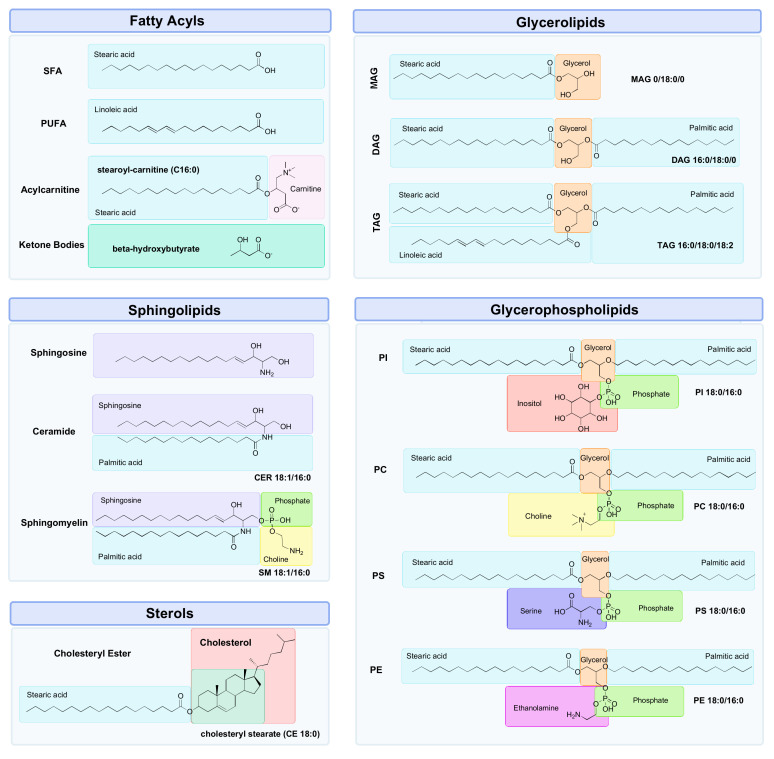
Different lipid classes most abundantly found in the brain. CE, cholesteryl ester; CER, ceramide; DAG, diacylglycerol; PC, phosphatidylcholine; PE, phosphatidylethanolamine; PI, phosphatidylinositol; PS, phosphatidylserine; MAG, monoacylglycerol; PUFA, polyunsaturated fatty acid; SFA, saturated fatty acid; SM, sphingomyelin; TAG, triacylglycerol.

**Table 1 metabolites-12-00822-t001:** Changes in fatty acyls with ageing, cognitive decline and exercise in humans.

Subclass	Species	Change with Ageing	Change with Cognitive Decline	Change with Exercise
Acylcarnitine	Acetylcarnitine (C2)	↑ plasma [81] ↑ serum [82] ↑ CSF [83]	↓ serum levels progressively decreased from CN > SMC > MCI > AD [84] ↓ serum levels associated with worse cognition and lower MMSE scores in AD [85]	↑ plasma [86,87,88,89]
	Propionylcarnitine (C3:0)	↑ plasma [81,90]	↓ baseline plasma levels in MCI/AD converters than CN [91] ↓ plasma levels progressively decreased from CN > Converter_pre_ > MCI/AD [92]	↓ plasma [86] ↑ plasma [88] ↑ serum [93]
	Malonylcarnitine (C3-DC)	↑ serum [82]	↑ baseline serum levels in CN and MCI associated with slower decline in SM [94]	
	Butyrylcarnitine (C4)	↑ plasma [81] ↑ serum [82]		↑ plasma [86,88] ↑ serum[93]
	Hexanoylcarnitine (C6)	↑ plasma [81] ↑ serum [82]	↓ serum levels associated with worse cognition and lower MMSE scores in AD [85]	↑ plasma [86,88,89] ↑ serum [93]
	Hexenoylcarnitine (C6:1)	↑ serum [82]	↓ serum levels progressively decreased from CN > SMC > MCI > AD [84] ↑ baseline serum levels in CN and MCI associated with reduced AD risk and slower decline in GC, EM and SM [94]	↑ plasma [88]
	Pimelylcarnitine (C7DC)	↓ plasma [90]	↑ baseline serum levels in CN and MCI associated with reduced AD risk and slower decline in GC, EM and SM [94]	↑ plasma [86]
	Octanoylcarnitine (C8:0)	↑ plasma [81,90,95] ↑ serum [82]	↑ serum levels in MCI and then decreased slightly in AD (CN < AD < MCI) [96] ↓ serum levels associated with worse cognition and lower MMSE scores in AD [85]	↑ plasma [86,88,97] ↑ serum [93]
	Nonanoylcarnitine (C9:0)	↓ plasma [90] ↑ serum [82]	↓ baseline plasma levels in MCI/AD converters than CN [91]	↑ plasma [88]
	Decanoylcarnitine (C10)	↑ plasma [81] ↑ serum [82]	↑ serum levels progressively increased from CN < MCI < AD [96] ↑ baseline serum levels in CN and MCI associated with reduced AD risk and slower decline in GC and SM [94] ↓ serum (CN > SMC > MCI > AD) [84] ↓ serum levels associated with worse cognition and lower MMSE scores in AD [85] ↑ plasma levels progressively increased from CN < MCI < AD [98]	↑ plasma [86,88,89,97]↑ serum [93]
	Decenoylcarnitine (C10:1)	↑ plasma [81] ↑ serum [82,99]	↑ baseline plasma levels in MCI/AD converters than CN [91] ↑ serum levels increased in AD but not MCI when compared to CN [96] ↓ serum levels associated with worse cognition and lower MMSE scores in AD [85] ↑ plasma levels progressively increased from CN < MCI < AD [98]	↑ plasma [86,88,97]
	Decadienoylcarnitine (C10:2)	↑ plasma [90,100] ↑ serum [82]	↓ baseline plasma levels in MCI/AD converters than CN [91]	↑ plasma [88]
	Dodecanoylcarnitine (C12)	↑ plasma [101] ↑ serum [82]	↓ serum levels progressively decreased from CN > SMC > MCI > AD [84]	↑ plasma [86,97]
	Myristoylcarnitine (C14:0)	↑ plasma [81,90,102] ↑ serum [82]	↓ serum levels progressively decreased from CN > SMC > MCI > AD [84]	↑ plasma [86,97]
	Tetradecenoyl- carnitine (C14:1)	↑ plasma [81,101,103,104] ↑ serum [82]	↓ serum levels progressively decreased from CN > SMC > MCI > AD [84]	↑ plasma [86]
	Tetradecadienoyl- carnitine (C14:2)	↑ plasma [81,90] ↑ serum [82]	↑ baseline serum levels in CN and MCI associated with reduced AD risk and slower decline in GC and SM [94]	
	Tetradecadien- carnitine (C14:2)		↓ serum levels associated with worse cognition and lower MMSE scores in AD [85]	
	Palmitoylcarnitine (C16:0)	↑ plasma [81,102] ↑ serum [82]	↑ serum levels in AD, but no change in MCI and CN [104]	↑ plasma [86,88,89]
	Hexadecenoyl- Hydroxy-carnitine (C16:1-OH)		↓ plasma levels progressively decreased from CN > Converter_pre_ > MCI/AD [92]	
	Hexadecenoyl- carnitine (C16:1)	↑ plasma [81,90,102]↑ serum [82]	↓ serum levels progressively decreased from CN > SMC > MCI > AD [84]	↑ plasma [86,88] ↑ serum [93]
	Hexadecadienoyl- carnitine (C16:2)	↑ plasma [90]	↑ baseline plasma levels in MCI/AD converters than CN [91]	↑ plasma [88]
	Octadecanoyl- carnitine (C18:0)	↑ plasma [81,102] ↑ serum [82,105]	↓ serum levels progressively decreased from CN > SMC > MCI > AD [84] ↑ serum levels in MCI and then decreased slightly in AD (CN < AD < MCI) [104]	↑ plasma [86]
	Octadecenoyl- carnitine (C18:1)	↑ plasma [81,90] ↑ serum [82,99]	↓ serum levels progressively decreased from CN > SMC > MCI > AD [84] ↑ serum levels in MCI and then decreased in AD (CN < AD < MCI) [104]	↑ plasma [86,89]
	Octadecadienyl- carnitine (C18:2)	↑ plasma [81]	↓ serum levels progressively decreased from CN > SMC > MCI > AD [84] ↑ serum levels in MCI and then decreased in AD (CN < AD < MCI) [104]	↑ plasma [86]
Ketone bodies	Beta-hydroxy butyrate	↑ plasma [106]	↓ levels in AD plasma [103] ↑ plasma levels in MCI improvements in cognitive function and positively correlated with plasma ketone levels [107]	↑ plasma [86,97] ↑ serum [93]
Saturated Fatty acids	Capric acid (C10:0)		↑ CSF levels increased from CN to MCI and then slightly decreased in AD (CN < AD < MCI) [108]	↑ plasma [86,88,97] ↑ serum [109]
	Undecylic acid (C11:0)		↑ CSF levels progressively increased from CN < MCI < AD[108]	
	Myristic acid (C14:0)		↑ serum levels in MCI > CN [110] ↑ erythrocyte levels in SMC [111] ↑ CSF levels progressively increased from CN < MCI < AD [108]	↑ plasma [86,88,97] ↑ serum [109]
	Pentadecylic acid (C15:0)		↑ CSF levels increased from CN to MCI and then slightly decreased in AD (CN < AD < MCI) [108]	
	Palmitic acid (C16:0)		↑ levels associated with increased risk of cognitive decline [112] ↑ serum levels in MCI > CN [110] ↓ plasma levels progressively decreased from CN > MCI > AD [113] ↑ CSF levels progressively increased from CN < MCI < AD [108] ↓ plasma levels progressively decreased from CN < MCI < AD [98]	↑ plasma [86,88,97]
	Margaric acid (C17:0)		↑ CSF levels increased from CN to MCI and then slightly decreased in AD (CN < AD < MCI) [108]	↑ plasma [86,88,97]
	Stearic acid (C18:0)	↑ plasma [106] plasma [114]	↓ plasma levels decreased from CN > MCI, then increased slightly in AD (CN > AD > MCI) [113] ↑ levels associated with greater risk of cognitive decline [115] ↑ CSF levels decreased from CN to MCI and then increased in AD (MCI < CN < AD) [108]	↑ plasma [86,97]
	Behenic acid (C22:0)		↓ serum levels in MCI > CN [110]	↑ plasma [86,97]
MUFA	Pentadecenoic acid (C15:1)		↑ CSF levels progressively increased from CN < MCI < AD [108]	
	Palmitoleic acid (C16:1)		↑ serum levels in MCI > CN [110] ↑ CSF levels increased from CN to MCI and then slightly decreased in AD (CN < AD < MCI) [108]	↑ plasma [86,88,97] ↑ serum [109]
	Heptadecenoic acid (C17:1)	↑ plasma[95]		↑ plasma [86,88,97]
	Oleic acid (C18:1)	↑ plasma [106]	↓ plasma levels progressively decreased from CN > MCI > AD [113] ↑ CSF levels increased from CN to MCI and then declined below CN levels in AD (AD < CN < MCI) [108] ↓ plasma levels progressively decreased from CN < MCI < AD [98]	↑ plasma [86,89,97] ↑ serum [109]
	Nonadecenoic acid (C19:1)		↑ CSF levels progressively increased from CN < MCI < AD [108]	↑ plasma [86,88,97]
	Nervonic acid (C24:1)		↓ serum levels in MCI > CN [110]	↑ plasma [86,88,97]
PUFA	Linoleic acid (C18:2)	↑ plasma [106] ↓ erythrocyte [111]	↓ plasma levels progressively decreased from CN > MCI > AD [113] ↑ plasma levels in MCI > CN [116] ↑ CSF levels increased from CN to MCI and then slightly decreased in AD (CN < AD < MCI) [108]	↑ plasma [86,89,97] ↑ serum [93]
	Linolenic acid (C18:3)		↑ CSF levels increased from CN to MCI and then slightly decreased in AD (CN < AD < MCI) [108]	↑ plasma [86,89,97]
	Eicosadienoic acid (C20:2n-6)	↓ plasma [95]	↑ erythrocyte levels associated with lower MMSE and executive function scores [117] ↑ CSF levels progressively increased from CN < MCI < AD [108] ↓ plasma levels decreased from CN < MCI then increased slightly in AD (CN > AD > MCI) [98]	↑ plasma [86,89,97]
	Eicosatrienoic acid (C20:3)		↑ CSF levels increased from CN to MCI and then slightly decreased in AD (CN < AD < MCI) [108] ↓ plasma levels decreased from CN < MCI then increased slightly in AD (CN > AD > MCI) [98]	↑ serum [86,89,97]
	Arachidonic acid [AA,(C20:4n-6)]	↑ plasma [106]	↑ levels associated with increased risk of cognitive decline [112] ↑ erythrocyte levels predicted cognitive impairment [117] ↑ CSF levels decreased from CN > MCI, then increased in AD (MCI < CN < AD) [108]	↑ plasma[86]
	Eicosapentanoic acid [EPA, (C20:5n-3)]	↓ plasma [118] ↑ plasma [95] ↑ CSF [119]	↓ levels associated with increased risk of cognitive decline [112] ↑ CSF levels increased from CN < MCI then declined below CN levels in AD (AD < CN < MCI) [108] ↑ serum levels positively associated with cognition [120] ↓ plasma levels progressively decreased from CN < MCI < AD [98]	↑ plasma [86] ↑ serum [93]
	Docosapentaenoic acid [DPA, (C22:5n-3)]	↑ CSF [119]	↑ CSF levels progressively increased from CN < MCI < AD [108] ↓ plasma levels decreased from CN < MCI then increased slightly in AD (CN > AD > MCI) [98]	↑ plasma [86,88,97] ↑ serum [93]
	Docosahexaenoic acid [DHA, (C22:6n-3)]	↑ CSF [119] ↓ brain [121] ↓ erythrocyte [111]	↓ levels associated with increased risk of cognitive decline [112] ↓ plasma levels decreased from CN > MCI then returned to normal levels in AD (CN = AD > MCI)[113] ↓ serum levels in MCI > CN [110] ↓ serum levels in MCI > CN [122] ↓ baseline levels in AD were associated with a higher risk of cognitive decline [123] ↑ blood levels associated with lower risk of AD and dementia [124] ↓ levels associated with declines in memory and executive function [125] ↑ CSF levels progressively increased from CN < MCI < AD [108] ↓ plasma levels progressively decreased from CN < MCI < AD [98]	↑ plasma [86,88] ↑ serum [93]

↑, increased levels; ↓, decreased levels; AD, Alzheimer’s disease; CN, cognitively normal; Converter_pre_, baseline samples from individuals who phenoconverted from CN to MCI or AD; CSF, cerebrospinal fluid; MCI, mild cognitive impairment; MMSE, Mini-Mental State Examination; MUFA, mono-unsaturated fatty acid; PUFA, poly-unsaturated fatty acid; SMC, subjective memory complaint; >, greater than; <, less than; =, no difference.

**Table 2 metabolites-12-00822-t002:** Changes in glycerolipids with ageing, cognitive decline and exercise in humans.

Subclass	Species	Change with Ageing	Change with Cognitive Decline	Change with Exercise
MAG	2AG	↑ plasma [168]		
DAG	DAG 16:0/18:1			↓ plasma [86,169]
	DAG 16:0/18:2			↓ plasma [86,169]
	DAG 16:1/18:2			↓ plasma [86,169]
	DAG 18:1/18:1			↓ plasma [86,169]
	DAG 18:1/18:2			↓ plasma [86,169]
TAG	TAG 48:1	↑ serum [170]		
	TAG 50:0	↑ serum [170]		
	TAG 52:1	↑ serum [170]		
	TAG 56:7	↓ plasma [100] ↑ CSF [119]	↓ plasma levels progressively decreased from CN > MCI > AD [98]	
	TAG 56:8	↓ plasma [101] ↑ CSF [119]	↓ plasma levels progressively decreased from CN > MCI > AD [98]	

↑, increased levels; ↓, decreased levels; AD, Alzheimer’s disease; CN, cognitively normal; CSF, cerebrospinal fluid; MCI, mild cognitive impairment; >, greater than; <, less than; MAG, monoacylglycerol; TAG, triacylglycerol; 2AG, 2-archidonoylglycerol.

**Table 3 metabolites-12-00822-t003:** Changes in phospholipids with ageing, cognitive decline and exercise in humans.

Subclass	Species	Change with Ageing	Change with Cognitive Decline	Change with Exercise
PC	Glycero- phosphocholine	↓ serum [173] ↑ plasma [106]	↓ serum levels associated with worse cognition and lower MMSE scores in AD [85]	
LysoPC	LPC a C18:2	↓ serum [196]	↓ plasma levels progressively decreased from CN > Converter_pre_ > MCI/AD [92]	
	LPC 20:5		↓ serum levels progressively declined in CN > MCI > AD [104]	
	LPC 22:6		↓ serum levels progressively declined in CN > MCI > AD [104]	
PC	PC 16:0/16:0	↑ CSF [174]	↑ serum levels in MCI and then decreased in AD (CN < AD < MCI) [104]	
	PPC 16:0/18:2		↓ serum levels progressively declined in CN > MCI > AD [104]	
	PC 16:0/18:2	↑ serum [173]	↑ serum levels in MCI and then decreased in AD (CN < AD < MCI) [104]	
	PC 16:1/22:6		↓ serum levels in MCI and then increased in AD (CN > AD > MCI) [104]	
	PC 18:1/20:4	↑ CSF [174]	↓ serum (AD) [197]	↑ plasma [169]
	PC ae C26:1		↑ baseline serum levels in CN and MCI associated with faster decline in SM [94]	
	PC aa 30:0	↑ serum [82]	↑ baseline serum levels in CN and MCI associated with faster decline in GC [94]	
	PC ae 30:0	↑ serum [82]	↑ baseline serum levels in CN and MCI associated with faster decline in GC and SM [94]	
	PC ae C34:0	↑ serum [82]	↑ baseline serum levels in CN and MCI associated with faster decline in GC, EM, PS and SM [94]	
	PC ae C34:1	↑ serum [82]	↑ baseline serum levels in CN and MCI associated with faster decline in GC [94]	
	PC ae C36:2	↑ serum [82]	↑ baseline serum levels in CN and MCI associated with faster decline in GC and SM [94]	
	PC O-36:4		↓ plasma levels decreased from CN > MCI then stayed the same for AD (CN > MCI = AD) [98]	
	PC 36:5	↑ CSF [174] ↓ plasma [100,198]	↓ plasma levels progressively decreased from CN < MCI < AD [98]	
	PC aa C36:5	↑ serum [82]	↑ baseline serum levels in CN and MCI associated with slower decline in PS [94]	
	PC aa 36:6	↑ serum [82]	↓ plasma levels progressively decreased from CN > Converter_pre_ > MCI/AD [92]	
	PC 37:6		↓ plasma levels decreased from CN < MCI then slightly increased in AD (CN > AD > MCI) [98]	
	PC aa 38:0	↑ serum [82]	↓ plasma levels progressively decreased from CN > Converter_pre_ > MCI/AD [92]	
	PC aa C38:3	↑ serum [82]	↑ baseline serum levels in CN and MCI associated with faster decline in SM [94]	
	PC aa 38:5		↓ plasma levels decreased from CN < MCI then slightly increased in AD (CN > AD > MCI) [98]	
	PC aa C38:5	↑ CSF [174] ↓ plasma [100] ↓ CSF [108]	↑ baseline serum levels in CN and MCI associated with slower decline in PS [94]	
	PC 38:6	↓ plasma [100] ↓ CSF [108]	↓ plasma levels decreased from CN < MCI, then slightly increased in AD (CN > AD > MCI) [98]	
	PC aa C40:1	↑ serum [82]	↓ plasma levels progressively decreased from CN > Converter_pre_ > MCI/AD [92]	
	PC aa C40:2	↑ serum [82]	↓ plasma levels progressively decreased from CN > Converter_pre_ > MCI/AD [92]	
	PC aa 40:5	↑ serum [82] ↓ serum [105]	↓ plasma levels decreased from CN < MCI, then slightly increased in AD (CN > AD > MCI) [98]	
	PC aa 40:6	↓ plasma [100]	↓ plasma levels progressively decreased from CN < MCI < AD [98]	
	PC aa C40:6	↑ serum [82]	↓ plasma levels progressively decreased from CN > Converter_pre_ > MCI/AD [92]	
	PC ae C40:6	↓ plasma [95]	↓ plasma levels progressively decreased from CN > Converter_pre_ > MCI/AD [92]	
	PC ae C44:4		↑ baseline serum levels in CN and MCI associated with faster decline in GC and EM [94]	
PE	PE 16:0/18:0		↑ serum levels in MCI and then decreased in AD (CN < AD < MCI) [104]	
	PE 36:4		↑ plasma levels increased from CN < MCI, then stayed the same in AD (CN < MCI = AD) [98]	
	PE 38:5		↑ plasma levels increased from CN < MCI, then decreased slightly in AD (CN < AD < MCI) [98]	
	PE 38:7		↓ plasma levels progressively decreased from CN > MCI > AD [98]	
	PE 40:6	↓ CSF[119]	↓ plasma levels decreased from CN > MCI, then the stayed same in AD (CN > MCI = AD) [98]	
LysoPE	LPE 18:0/0:0		↑ plasma levels progressively increased from CN < MCI < AD [98]	
	LPE 18:1/0:0		↑ plasma levels increased from CN < MCI, then decreased slightly in AD (CN < AD < MCI) [98]	
PI	PI 40:6		↑ plasma levels decreased from CN > MCI, then increased slightly in AD (CN > AD > MCI) [98]	
LysoPI	LPI 18:0/0:0		↑ plasma levels positively associated with cognition [120]	

↑, increased levels; ↓, decreased levels; AD, Alzheimer’s disease; CN, cognitively normal; Converter_pred_, baseline samples from individuals who phenoconverted from CN to MCI or AD; CSF, cerebrospinal fluid; EF; executive function; EM, episodic memory; GC, global cognition; MCI, mild cognitive impairment; MMSE, Mini-Mental State Examination; PC; phosphatidylcholine; PE, phosphatidylethanolamine; PI, phosphatidylinositol; PS, perceptual speed; SM, semantic memory; SMC, subjective memory complaint; >, greater than; <, less than; =, no change.

**Table 4 metabolites-12-00822-t004:** Changes in sphingolipids with ageing, cognitive decline and exercise in humans.

Subclass	Species	Change with Ageing	Change with Cognitive Decline	Change with Exercise
Ceramides	Hex-CER 18:1/16:0		↑ serum levels in MCI and then decreased in AD (CN < AD < MCI) [104]	
	Hex-CER 18:1/18:0		↑ serum levels in MCI and then decreased in AD (CN < AD < MCI) [104]	
	Hex-CER 18:1/24:1(2OH)		↑ plasma levels negatively associated with cognition [120]	
	Lac-CER 18:1/14:0		↑ serum levels in MCI and then decreased in AD (CN < AD < MCI) [104]	
	Lac-CER 18:1/16:1		↑ serum levels in MCI and then decreased in AD (CN < AD < MCI) [104]	
	Lac-CER 18:1/16:0		↑ serum levels in MCI and then decreased in AD (CN < AD < MCI) [104]	
	CER 18:1/16:0		↑ serum levels progressively increased from CN < MCI < AD [104]	
	CER 39:1		↑ plasma levels decreased from CN > MCI, then increased slightly in AD (CN > AD > MCI) [98]	
	CER 40:1		↑ plasma levels decreased from CN > MCI, then increased slightly in AD (CN > AD > MCI) [98]	
	CER 41:1		↓ plasma levels progressively decreased from CN > MCI > AD [98]	
	CER 42:1		↑ plasma levels decreased from CN > MCI, then increased slightly in AD (CN > AD > MCI) [98]	
	CER 43:1		↓ plasma levels progressively decreased from CN > MCI > AD [98]	
Sphingo- myelins	SM 18:1/14:0	↑ plasma [95]	↑ serum levels in MCI and then decreased in AD (CN < AD < MCI) [104]	↓ plasma [86]
	SM 18:1/16:0	↑ plasma [95]	↑ serum levels in MCI and then decreased in AD (CN < AD < MCI) [104] ↓ CSF (MCI) [207]	↑ plasma [86] ↓ plasma [86]
	SM 18:1/18:1		↑ serum levels in MCI and then decreased in AD (CN < AD < MCI) [104]	↑ plasma [86]
	SM 18:1/18:0		↑ serum levels in MCI and then decreased in AD (CN < AD < MCI) [104]	↑ plasma [86] ↓ plasma [86]
	SM 18:1/22:0	↓ plasma [101]	↓ CSF (MCI) [207]	↑ plasma [86]
	SM 39:1	↓ plasma [100] ↓ CSF [119]	↓ plasma levels progressively decreased from CN > MCI > AD [98]	
	SM 41:1	↓ plasma [100]	↓ plasma levels progressively decreased from CN > MCI > AD [98]	↑ serum [208]
	SM 42:1	↓ plasma [100]	↓ plasma levels progressively decreased from CN > MCI > AD [98]	

↑, increased levels; ↓, decreased levels; AD, Alzheimer’s disease; CER; ceramide, Hex-CER, hexosyl ceramide; lac-CER, lactosyl ceramide; CN, cognitively normal; CSF, cerebrospinal fluid; MCI, mild cognitive impairment; SM, sphingomyelin >, greater than; <, less than; =, no change.

**Table 5 metabolites-12-00822-t005:** Changes in sterols with ageing, cognitive decline and exercise.

Subclass	Species	Change with Ageing	Change with Cognitive Decline	Change with Exercise
Sterols	Cholesterol	↑ plasma [106,200] ↑ plasma (F) [199]	↓ brain (AD) [222]	↓ plasma [86]
Cholesteryl esters	CE 20:3		↑ levels associated with increased risk of global cognitive decline [112]	
	CE 18:2	↓ plasma [100]	↓ levels associated with increased risk of global cognitive decline [112]	
Steroids	Dehydro- epiandrosterone-sulfate (DHEA-S)	↓ plasma [106]	Positive correlation between DHEA-S and global cognition in M and F. Positive correlations between DHEA-S on working memory, attention and verbal fluency in F only [223]	

↑, increased levels; ↓, decreased levels; AD, Alzheimer’s disease; CE; cholesteryl ester; DHEA-S, dehydroepiandrosterone sulfate; F, female; M, male.

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
