# Peer review of "Associations of the Lipidome with Ageing, Cognitive Decline and Exercise Behaviours"

_metabolites, 2022, doi:10.3390/metabo12090822_

Round 1
Reviewer 1 Report
The manuscript "Associations of the lipidome with ageing, cognitive decline and exercise behaviours" by Kadyrov et al. reviews the current literature regarding the associations of different lipids levels with aging, cognitive abilities, and exercise. The manuscript is comprehensive and well-structured. It can be very thought-provoking for readers of Metabolites, especially those interested in diagnosis and therapy of age-related cognitive decline.
The reviewer has only three minor suggestions, mainly related to the formatting of the text:
1. Please revise the numbering of paragraphs, as it is not always consistent (in lines 623, 779, 853, 863, 894, and 930).
2. Please revise the formatting of DOI numbers in the References section. Some references have pure DOI numbers, but others have them in https://doi.org/ format.
3. Figure 1 contains "Inflammation" listed twice (redundantly).
Author Response
Thank you for your report and valuable comments. We have made the following changes to our manuscript:
- We have revised the numbering of paragraphs on lines 623, 779, 853, 863, 894, and 930 to ensure continuity throughout the manuscript.
- We have revised the formatting of DOI numbers in the References section to remove the hyperlinks so that only the pure DOI numbers are listed.
- We have edited Figure 1 to remove the repeated word “inflammation”.
Reviewer 2 Report
This is an excellent review I have no specific critiques or suggestions for improvement
The authors comprehensively review the current state of the art of lipid metabolomics with specific attention towards alterations with aging and the development of cognitive decline as well as potential remediation with exercise. The paper seems comprehensively researched and seems to be very well written. I do not detect any particular bias or unjustified influence in this paper.
The presentation of lipid alterations with cognitive decline and aging as well as changes with exercise is as clear as any I have previously seen presented, with the caveat that the field generally suffers from a somewhat esoteric nomenclature, an unavoidable necessity for describing biological entities in the language of organic chemistry. However, without this seemingly necessary complexity the problems of incomplete or unclear nomenclature do multiply as the authors suggest. The authors provide thought-provoking commentary specifically related to the problems inherent with nomenclature discrepancies of specific lipid moieties in this field.
Overall it is my opinion that this paper should be of broad interest to both clinicians and basic scientists and to the readership of metabolites more generally. thank you for the opportunity to review this excellent manuscript.
Author Response
Thank you for your supportive report and encouraging words. We appreciate the time you have dedicated to reviewing our paper.